# Wearable Solutions Using Physiological Signals for Stress Monitoring on Individuals with Autism Spectrum Disorder (ASD): A Systematic Literature Review

**DOI:** 10.3390/s24248137

**Published:** 2024-12-20

**Authors:** Sandra Cano, Claudio Cubillos, Rodrigo Alfaro, Andrés Romo, Matías García, Fernando Moreira

**Affiliations:** 1School of Informatic Engineering, Pontificia Universidad Católica de Valparaíso, Valparaíso 2340000, Chile; sandra.cano@pucv.cl (S.C.); claudio.cubillos@pucv.cl (C.C.); rodrigo.alfaro@pucv.cl (R.A.); andres.romo.m@mail.pucv.cl (A.R.); matias.garcia@pucv.cl (M.G.); 2REMIT (Research on Economics, Management and Information Technologies), IJP (Instituto Jurídico Portucalense), Universidade Portucalense, Rua Dr. António Bernardino de Almeida, 541-619, 4200-072 Porto, Portugal; 3IEETA (Instituto de Engenharia Electrónica e Telemática de Aveiro), Universidade de Aveiro, 3810-193 Aveiro, Portugal

**Keywords:** stress detection, emotion detection, wearable technology, autism spectrum disorder, physiological signals, biomedical sensors

## Abstract

Some previous studies have focused on using physiological signals to detect stress in individuals with ASD through wearable devices, yet few have focused on how to design such solutions. Wearable technology may be a valuable tool to aid parents and caregivers in monitoring the emotional states of individuals with ASD who are at high risk of experiencing very stressful situations. However, effective wearable devices for individuals with ASD may need to differ from solutions for those without ASD. People with ASD often have sensory sensitivity and may, therefore, not tolerate certain types of accessories and experience discomfort when using them. We used the Scopus, PubMed, WoS, and IEEE-Xplore databases to search for studies published from 2014 to 2024 to answer four research questions related to wearable solutions for individuals with ASD, physiological parameters, and techniques/processes used for stress detection. Our review found 31 articles; not all studies considered individuals with ASD, and some were beyond the scope of this review. Most of the studies reviewed are based on cardiac activity for stress monitoring using photoplethysmography (PPG) and electrocardiography (ECG). However, limitations include small sample sizes, variability in study conditions, and the need for customization in stress detection algorithms. In addition, there is a need to customize the stress threshold due to the device’s high individual variability and sensitivity. The potential of wearable solutions for stress monitoring in ASD is evident, but challenges include the need for user-friendly and unobtrusive designs and integrating these technologies into comprehensive care plans.

## 1. Introduction

The prevalence of autism spectrum disorder has been steadily increasing, and with it, so has the need for effective strategies to support individuals with this condition. One key aspect of this support is the ability to detect and manage stress, which can have significant implications for the well-being and functioning of individuals with ASD. According to the *Diagnostic and Statistical Manual of Mental Disorders* (DMS-5) [1], people with ASD have difficulties in communication and social interaction because of atypical information processing and sensory integration abnormalities. These can cause a state of cognitive and emotional overload related to increased stress, which can lead to inappropriate social behaviors.

Stress affects people with ASD, children [2], adolescents [3], and adults [4]. High-stress levels may significantly impact their well-being and quality of life [5]. Studies indicate that individuals with ASD experience significantly higher levels of stress compared to neurotypical individuals [6].

Stress can have an impact on the physical and mental health of a person with ASD [7], as well as their overall functioning. High levels of stress can exacerbate existing symptoms, such as difficulty with social communication, rigid behaviors, and sensory sensitivities. Chronic stress can also lead to the development of secondary mental health conditions, such as depression and anxiety [8]. A study presented by Hollocks et al. [9] shows that individuals with ASD have higher rates of anxiety and depression compared to the general population.

People with ASD have problems related to sensory processing and can, therefore, experience a sensory overload, in which one or more senses react to stimuli, which can trigger elevated stress levels. Common stress triggers for individuals with ASD include sensitivity to sound and light [10]. Technological interventions, such as wearable stress sensors and mobile applications, show promise in monitoring stress reactivity and alerting caregivers [11].

Stress is a complex physiological and psychological response to environmental demands or challenges [12]. It can manifest as eustress (positive) or distress (negative) and is experienced through stimulus, response, and transactional models [13]. Chronic stress can lead to anxiety and depression [14], affecting multiple bodily systems, immune, cardiovascular, and neuroendocrine [15]. Therefore, stress is a significant issue for individuals with ASD, as they often experience heightened sensory sensitivities and difficulty regulating their emotional responses [16]. Monitoring and managing stress levels is crucial for this population, as chronic stress can lead to various negative health outcomes. Wearable technology offers a promising solution for real-time stress detection and intervention, allowing for continuous monitoring and personalized support [17].

Recent studies have explored the potential of utilizing wearable sensors to measure physiological indicators of stress in people with ASD [18], such as heart rate variability (HRV), heart rate (HR), electrodermal (EDA), respiration rate (RR), and blood pressure (BP). Heart rate variability calculates the time intervals between two consecutive R peaks in the ECG signal, if measured from ECG sensors. HRV reflects the autonomic regulation of the cardiovascular system, has been shown to be altered in individuals with ASD, and can provide insight into their stress [19]. Heart rate is a measure of the number of beats per minute but has limitations about the heart activity. Electrodermal activity reflects changes in sweat gland activity and may be a useful indicator of emotional arousal and stress responses in individuals with ASD [20]. Respiration rate is the total number of breaths, or respiratory cycles, that occur each minute, which has been shown as a useful indicator for stress [21]. These physiological measures can be used in conjunction with behavioral observations to obtain a comprehensive understanding of stress in individuals with ASD and their families. These wearable technologies aim to provide early warning of impending anxiety attacks or challenging behaviors, enabling timely interventions by caregivers and potentially improving stress management for individuals with ASD [22].

Research in this field has, therefore, been increasing. A review by Tal-Eldin [23] focused on wearable devices for monitoring physiological signals to be used by people with ASD, finding 25 commercial wearable devices that varied in form (i.e., wristbands, chest straps, vests, garments and shirts, patches, and sleeves), materials, sensors, and parameters that were designed for people without ASD but may be adapted for people with ASD. However, some devices are limited in terms of the number of sensors, data reading, data collection, and data access and have not yet been validated. Most of the reviewed devices are designed for the general population. The E4 wristband developed by Empatica (USA) has gained significant attention for its ability to capture a wide range of biometric data. The E4 wristband can measure vital signs such as heart rate, body temperature, and blood pressure, as well as physical activity and other behaviors [24]. However, to access the data, it is necessary to pay. A prototype called AutiSense [25] was developed as a wearable-technology-type glove that measures galvanic skin response (GSR) and heart rate (HR) and tracks HR variability. However, this device has not been validated in individuals with ASD. Another prototype proposed is Snap [26], a type of wristband that involves two functions: (1) alerts based on the automatic capture of physiological data and (2) tactile interaction (through the use of bubble wrap, tactile jewelry, etc.), which measures changes in the electrical resistance of the body. For this device, a preliminary test was carried out on adults with ASD. Black et al. [27] published a literature review about the use of wearable technology centered on autistic youth. This technology, based on accessories and clothes, included wrist-worn devices, sensorized shirts, and devices worn around the neck. The reviewed studies reported that the esthetics of the devices were a concern for the participants, suggesting that the researchers did not consider aspects such as the size or appearance of the wearable device. This is particularly relevant in ASD, where the use of devices that are socially disruptive may further exacerbate social difficulties [28]. Finally, the authors of [27,29] presented a few studies related to the use of smart wearables for individuals with ASD. In addition, the use of machine learning (ML) approaches to predict aggressive behaviors using accelerometers and inertial measurement units has increased [30], employing techniques such as support vector machine (SVM), k nearest neighbor (kNN), decision tree (DT), and deep neural networks. The authors suggest that repetitive behavior can be effectively detected using wearable sensors.

The growth of wearable technology, meanwhile, has created research interest in monitoring stress in people with ASD using biomedical sensors, thereby allowing researchers to capture a person’s physiological responses. Stress response involves a complex interplay between physiological, psychological, and behavioral processes, which can vary across different situations [31]. The research has shown that stress can have both positive and negative effects on an individual’s emotional state. However, excessive or prolonged stress can lead to the manifestation of physical and mental health problems, such as anxiety, depression, and burnout [32]. The stress response is accompanied by a number of physiological changes, including increased heart rate, blood pressure, and muscle tension [31]. These physiological signals can provide important information about an individual’s stress levels and emotional state. Therefore, monitoring such responses in people with ASD can serve as a tool for their caregivers or parents that provides information about changes to their internal emotional states, as well as allow these individuals themselves to understand in real time the emotional changes they are experiencing. Many individuals with ASD struggle to understand and recognize their own emotions, making it difficult for them to notice that they are experiencing stress [33]. Researchers have explored the use of wearable devices such as watches, wristbands and bracelets, which seem to be the preferred type of wearable for people with ASD. However, not all individuals with ASD tolerate the same type of wearable device. In a study conducted by Goodwin et al. [34], they captured physiological signals using a wrist-worn biosensor to predict aggression in individuals with ASD. In that study, 20 youths with ASD tolerated the E4 device well. Watches/wristbands and bracelets have been found to be the most preferred wearable technology types [25].

Studies have shown promising results in their experiments; the real-world applicability of these methods remains uncertain since traditional machine learning models are only optimized for the specific dataset and are hard to personalize for individual users. In addition, the development of machine-learning-based stress detection systems is the acquisition of high-quality training data. Therefore, the complexity and high dimensionality of physiological data present significant challenges for conventional data analysis methods [35]. Wearable devices are growing in popularity, while machine learning models still require further study and research in this area.

This article is structured as follows: Section 2 defines the theoretical concepts related to stress and the physiological signals for stress monitoring. Section 3 presents the objectives of this systematic literature review and defines its research questions. Section 4 explains the methodology used for the literature search. Section 5 provides the results found in the articles selected. Section 6 discusses the reviewed studies. Finally, Section 7 offers the conclusions.

## 2. Background

### 2.1. Stress

Hans Selye defined stress as “the non-specific response of the body to any demand” [36]. Stress is associated with several disorders and related health problems. Therefore, in stressful situations, the body can respond with a number of neurohormonal changes, which depend on the activation of the hypothalamic–pituitary axis, neurons, the adrenomedullary system, and the parasympathetic system. The DSM-5 mentions that psychological distress associated with stress and trauma is varied and may include anxiety, changes in mood, anger, aggression, or dissociation. Therefore, stress is identified as a risk factor for several other disorders, including depression and [1]. Stress for individuals with ASD can stem from a variety of factors, including sensory processing difficulties, social communication challenges, and the need for routines and predictability [37], which can lead to significant distress and discomfort. Stress can also affect learning and motivation [38,39].

People with ASD face difficulties in sensory processing in hearing, vision, touch, taste, and smell [40], such as bright lights or noisy sounds [40], which is due to their hypo- or hyperarousal to sensory input. Overall, hypersensitive people face major difficulties and experience distress when presented with sensory stimuli. Tomchek and Dunn [40] purposed a comparative study to investigate differences in sensory processing in children between ages 3 and 6 years with and without ASD, where it was reported to have difficulties with processing and responding to sensory input. One of the hallmark features of autism is atypical sensory processing, with as many as 90% of individuals with autism exhibiting unusual responses to sensory information [41]. This diminished sensory sensitivity can have a significant impact on an individual’s daily functioning and overall well-being. Individuals with autism often experience difficulties in this domain, leading to behaviors such as hypersensitivity to certain stimuli, sensory-seeking behaviors, or a combination of both. These sensory processing challenges can contribute to increased stress and anxiety as the individual struggles to regulate their responses to sensory input [42].

The recent research has explored the relationship between sensory processing and stress in individuals with autism. The findings suggest that sensory problems, such as sensitivity to specific sounds, textures, or lights, are a significant predictor of parenting stress in families of children with autism. Children with autism who exhibit more pronounced sensory difficulties tend to have parents who report higher levels of stress, which can, in turn, impact the overall family dynamics and functioning [3].

Individuals with ASD have problems with stress management due to their sensory processing issues [43]. On an emotional level, sensory processing sensitivity is associated with high anxiety levels [34], negative emotions [44,45], and distress [46]. Several studies have shown that sensory processing difficulties have a negative impact on well-being and quality of life [47]. Another study [48] indicates that sensory processing issues may interfere more for individuals with hypersensitiveness, because they may experience more stress. Therefore, hypersensitiveness is a relevant factor for stress in individuals with autism. Individuals with ASD suffer during moments of crisis due to their inability to control their emotions. These manifestations may have very dangerous consequences for them, especially in cases of aggressive behavior [49].

Stress is associated with increases in arousal, i.e., when there is stress, our heart rate, breathing, and blood pressure increase and our body begins to secrete adrenaline and other hormones [49]. Stress can be manifested through changes in physiological responses such as electrodermal activity (EDA) [50], heart activity [51], respiration activity [52], and skin temperature (ST) [53,54].

Most published research on individuals with ASD uses questionnaires or occasionally saliva biomarkers [55] but rarely utilizes direct and continuous assessment. Stress in people with ASD needs continuous monitoring to enable timely intervention by caregivers, and this can be achieved with the measurement of physiological signals.

### 2.2. Physiological Signals

The physiological responses used in stress detection include electroencephalogram (EEG) [56], cardiac activity [57], electromyogram (EMG) [58], electrodermal activity (EDA) [59], skin temperature [23], respiration, and IMU [60].

Cardiac activity is associated with the heartbeat, involving measurements such as heart rate [61] (HR) and heart rate variability (HRV). Cardiac activity can be captured using a photoplethysmography (PPG) sensor [58] or electrocardiograph (ECG) [62]. ECG is a recording of the electrical activity of the heart. PPG is a device that contains a light source and a photodetector. The light source emits light to a tissue and the photodetector measures the light reflected from the tissue. The reflected light is proportional to blood volume variation [63]. Bolaños et al. [63] compared the HRV signals extracted from PPG and ECG and demonstrated that PPG has excellent potential to replace ECG recordings for the extraction of HRV signals. PPG is the most commonly used sensor on wearable devices, with a green LED as the main light source. PPG must be used on specific body parts and each body part location has a different degree of accuracy. Body parts often used include the finger, earlobe, forehead, and forearm/wrist. HRV is the variation between two consecutive beats; therefore, a high HRV reflects the fact that an individual can constantly adapt to micro-environmental changes, while a decrease in HRV reflects a high stress level [64]. The resting HR of a person generally ranges from 60 to 90 beats per minute. When a person becomes stressed, their blood pressure increases, causing an increase in HR, which is linked with a low HRV [65].

Electrodermal activity (EDA), also called galvanic skin response (GSR), measures the conductivity of the skin and can be a useful indicator of stress [66]. GSR is measured using a pair of electrodes placed on the surface of the skin, where one electrode injects an alternating current with a small amplitude into the skin and the other is used to calculate the impedance of the skin using Ohm’s Law given a certain voltage. GSR consists of two main components: skin conductance level (SCL) and skin conductance response (SCR). SCL changes slightly on a time scale of tens of seconds to minutes. There are studies that propose methods for detecting stress levels by measuring skin conductance [67]. SCR is a symptom indicator in mental disorders; patients with depressive disorders, for example, exhibit low SCR [68]. Another study [69] showed that GSR is related to latency time and rise time; when a participant is happy, it indicates a longer recovery time, but when they are experiencing fear, the rise time appears shorter. Finally, when the participant is experiencing happiness, the response appears to be stable. EDA mean levels usually range between 2 and 20 μS and vary within a range between 1 and 3 μS for different individuals [21]. The authors extracted three Hjorth features, namely, activity, mobility, and complexity. They also extracted features in the time domain (event, statistical, and Hjorth), frequency domain (discrete wavelet transform and stationary wavelet transform), and time–frequency domain (MFCC), implementing 40 different EDA features.

Skin temperature (ST) is increased when individuals are exposed to emotional events. Emotions and temperature are related through embodiment [53]. A number of studies indicate that skin temperature increases in the presence of stress and decreases when there is a low level of stress [70]. Noelke et al. [70] conducted a comparative study with an average daily ST of 10–16 °C; temperatures above 21 °C—especially above 32 °C—significantly reduce positive and increase negative emotions. ST requires skin contact; changes in ST may help for identifying stress levels. However, whether or not it is useful for stress detection depends very much on the location of the measurement.

Respiration is related to cardiovascular system activity and is influenced by changes between calm and excited states. Respiration signals are used as physiological measures of states of anxiety [71]. An increase in respiration rate is associated with an increase in stress levels [72]. There are methods to measure respiration that can be classified into four approaches: manual measurements [73], measuring changes in impedance using electrodes [74], measuring pressure using belt-type sensors [75], and measuring airflow from the nose and mouth [76]. However, these approaches share several limitations, such as the fact that sets of special equipment are necessary for each, which are not usually available. In addition, the preparation to perform the measurement is long and complex.

As the field of stress detection continues to evolve, the integration of multiple physiological signals and the development of personalized stress detection algorithms will be crucial for the development of effective and reliable stress management solutions [21]. A study by Zhang et al. [77] proposed a predictive multimodal framework to alert caregivers of problem behaviors for individuals with ASD. Four physiological signals were used, namely, cardiac activity, EDA, body temperature, and three-axis acceleration, and were collected through the E4 wristband. Furthermore, a body movement tracking sensor was designed for measuring upper body motion using inertial measurement units (IMUs). BVP and EDA were collected with sampling rates of 64 Hz and 4 Hz, and BVP signals were filtered using a band-pass filter. The acceleration signals of the E4 were collected at 32 Hz, and a low-pass filter was applied. The features combined were heart rate and skin conductance, three-axis acceleration, and 20 roll, pitch, yaw, and activity levels from forearms, upper arms, and torso using IMUs. Kim K.H [78] found patterns between sadness, anger, stress, and surprise using signals representing heart activity, skin conductance, and temperature.

The studies reviewed above show the importance of real-time stress detection for various applications from workplace productivity to mental health monitoring [4]. Detecting stress in the user, particularly individuals with ASD who often experience heightened levels of stress and difficulty with emotional self-regulation, is important [79]. This can lead to a range of challenging behaviors, such as aggression, self-harm, and social isolation, which can significantly impact their quality of life and that of their parents or caregivers. Therefore, wearables can provide real-time feedback and assist individuals with ASD in implementing customized self-regulation strategies, leading to improved emotional control and reduced instances of stress-induced behaviors. These devices integrate multiple sensors, processors, and communication technologies to capture and transmit data, allowing for real-time assessment of an individual’s physiological state as stress levels. For example, wearable devices can monitor physiological indicators of arousal, such as heart rate or skin conductance, and provide discreet feedback or prompts to help caregivers for timely interventions.

## 3. Objectives

The objective of this systematic literature review is to examine the research that has been carried out on wearable technology used to detect stress in people with ASD.

The objective is to provide a synthesis of the current research to increase our understanding of the types of sensors and wearables used by people with ASD and the purposes for which they are used.

This systematic review aims to address the following research questions:What kind of wearable solutions are most acceptable for people with ASD?What features extraction/parameters are used for stress detection?What techniques/processes are used for stress detection in individuals with ASD?What techniques are applied for multimodal physiological signal fusion?

## 4. Methodology

This review was conducted via a systematic search of the available literature published in the past 10 years and was carried out according to the Preferred Reporting Items for Systematic reviews and Meta-Analysis (PRISMA) guidelines [70].

### 4.1. Study Selection Process

Due to the range of studies considering the use of wearable technology for individuals with ASD, it was decided to narrow the scope of the review to only include studies that consider physiological signals for individuals with ASD. Therefore, we applied a model commonly used for clinical applications and healthcare based on participants, interventions, comparison, outcomes, and study design (PICOS) framework:Participants: Only individuals with ASD children and adolescents.Interventions: Wearable sensors for detecting stress.Comparisons: The measured physiological signals for detection of stress or emotions.Outcome: qualitative and quantitative studies, excluding other systematic literature reviews or meta-analyses.Study Design: Observational and experimental studies.

### 4.2. Search Strategy

The Scopus, PubMed, Web of Science, and IEEE Xplore databases were searched. The search string used: (“wearable” OR “device” OR “wearable technology” OR “smart” OR “wireless” OR “IoT”) AND (“autism “ OR “asd”) and (“stress” or “anxiety” or “emotions”).

### 4.3. Screening and Eligibility

The inclusion and exclusion criteria were determined prior to conducting the searches. The articles that were included in this review were (1) articles from disciplines from journal title and keywords and abstracts related to computer science, stress, anxiety, and wearable technology and (2) only articles, lectures, and book chapters. The excluded articles were (1) those not available in English, (2) literature reviews, (3) articles unrelated to the purpose of the study, and (4) duplicate articles. The search or selection process was limited to specific disciplines, because the study is centered in topics related to signal processing and sensors used for monitoring physiological signals.

## 5. Results

The search of the databases resulted in a total of 31 articles. Finally, 31 articles published between 2014 and 2024 were selected after considering the inclusion and exclusion criteria. The selected articles allowed us to answer the study questions.

### 5.1. Data Synthesis

Figure 1 presents the flow diagram following PRISMA guidelines, and Table 1 shows the abstracted data, including study characteristics (year, population), intervention details (type of wearable sensors), outcomes measured (stress levels), purpose, and quality assessment (signals/parameters).

Figure 2 shows studies published by year considering a range of 10 years. Studies increase as of 2019.

The analyzed articles can be divided into three categories: conference articles, comprising 15 papers; journal papers, comprising 26 articles; and finally, book chapters, comprising 1 document. These proportions are shown in Figure 3.

The VOSviewer (Version 1.6.19) tool for network-based visualization was used, along with the Scopus and WoS databases, which show research trends and clusters of topics related to Autism. Figure 4 shows trends that are low with wearable technology, where the physiological signal most used is heart activity.

### 5.2. Answering the Research Questions

The objective of the abstraction was to respond to the following questions:

**Question** **1.**
*What kind of wearable solutions are most acceptable for people with ASD?*


In a first study [80], physiological data were taken with a chest strap electrocardiographic sensor in a group of 20 people diagnosed with ASD: 12 children, 6 adolescents and 2 adults. The authors designed an evaluation protocol that consisted of a first stage of relaxation for 7 min, where children viewed a relaxing video. Then, the children engaged in two tasks designed to mimic stressful scenarios. These tasks were based on the temperament assessment battery [81] for 5 min.

A second study [57] used 3 types of sensors for arousal detection, such as (1) Affectiva Q sensor, a Bluetooth-enabled sensor that collects data such as GSR and ST; (2) MindWave Mobile, a Bluetooth device to capture EEG power spectrums (alpha waves and beta); (3) Zephyr BioHarness, a Bluetooth sensor to capture HR, HRV, RT, ST, and movement. So, using many devices on the body can be invasive. However, the author attempted to demonstrate that the use of sensors and mobile technologies has potential to assist caregivers and parents in the care of children. Another study [82] used a shirt incorporating ECG and ACC sensors and a Bluetooth module. ECG signals are measured at the wrist as the rubber presses the electrode to the skin. Authors evaluated 4 children with ASC for the acceptability of the developed device during therapeutic sessions. Three of the four children agreed to wear the developed device for 60 min. However, when wearing a wearable type of garment, it is more challenging, as the size can change depending on the age or physical build of the child.

Pittella et al. [82] proposed a thoracic belt system, where they designed a sensor for respiratory and cardiac monitoring through a piezoelectric sensor and ECG. The authors evaluated how heart rate variability and breath signal are useful indicators of the stress response in children.

In a study proposed by the authors of [83], skin resistance was evaluated in four placements on the body such as on the arm and on the leg: 1—index and middle fingers (reference point), 2—wrist, 3—forearm, and 4—ankle. The authors found the values obtained on the ankle varied most; while those on the forearm had the narrowest distribution, similar to the wrist; and the data measured in the middle and index fingers affirmed this point as a reference. However, the resistance levels suggested more accurate measurements on the wrist.

The studies reviewed used commercially available devices such as Mio Fuse, Polar H7 chest [84], PulseON, Samsung Galaxy 3 [85], Hexoskin [86], Empatica E4 [87,88], LG Watch urbane [89], and Shimmer ECG [89]. Ref. [83] evaluated commercial devices for measuring and transmitting real-time physiological responses in children with ASD. However, the authors mention some limitations such as (1) the devices were not designed to measure accelerometry simultaneously; (2) the cardiovascular monitors evaluated found low robustness on some devices, which may have been due to some Bluetooth/ANT + interference across; and (3) due to the commercial nature of these devices, there is no access to the proprietary algorithms that pre-processed the heart rate/heart rate variability data.

Finally, the studies reviewed used mostly the belt and wrist, which have garnered significant attention due to their ability to unobtrusively track a range of physiological and behavioral data. In addition, most of the studies are conducted with male individuals, and in very few of them has the user experience been evaluated. In addition, very few describe the evaluation protocol, and in many of them, the evaluation is related to weeks of use and data collection.

Most of the studies focused on analyzing physiological cues in the behavior of children with ASD rather than technology acceptance. A study [88] designed an experiment to evaluate the acceptability of the developed device during therapeutic sessions, where 7 tasks were designed with an approximate duration of 60 min. The behaviors exhibited during the session were recorded by video cameras. Another study [90] mentions that the use of smartwatches does not make any difference in a community where other individuals also wear wearable. The difference may well be at the software level, but the implicit interaction of users with ASD and wearables is yet to be more broadly explored.

Finally, Table 1 shows a summary of the published research on wearable solutions for stress monitoring, where it shows different wearables and sensors used and techniques applied for feature extraction and study objective and participants.

**Question** **2.**
*What feature extraction/parameters are used for stress detection?*


In the reviewed studies, heart activity, EDA, ST, respiration rate (RR), and movement were measured. To capture the heart activity, there are two methods, namely, ECG and PPG. An electrocardiogram requires two electrode pads worn on the chest. Some features measured by ECG are as follows: HRV, SDNN, RMSSD, pNN50, VLF, LF, HF, QRS-complex, and P and T waves. The disadvantage is that the stickiness of the ECG electrodes may deteriorate over the course of the day, resulting in multiple inaccurate readings [83].

In ref. [91], RMSSD is a measure of HRV in the time domain, and it is recommended due to its superior statistical properties. However, the ECG sensor was the most used in the studies reviewed [82,92,93]. Another sensor was PPG, which is an optical measure of the arterial pulse wave, which is influenced by the heart and blood vessels, while ECG directly measures the electrical activity of the heart. PPG was used in many of the stress-related studies found (see Table 1), and this is due to the simplicity of attaching the sensors. In addition, PPG can be obtained using a wide range of devices, such as smartphones, tablets, and fitness devices. Some PPG features used in the studies found were the mean of pulse rates of a time segment, the standard deviation of pulse rates of a time segment, the variance of pulse rates of a time segment, the absolute PSD of the LF of HRV, the absolute PSD of the HF of HRV, and the absolute PSD of the LF/HF of HRV [94,95]. HRV has been associated with anxiety [96] and stress [97]. The authors of [91] concluded that HR increase can predict challenging behavior episodes in preschoolers with autism. Therefore, they suggest that HR monitoring may be useful in moment-to-moment decision support for the prevention of challenging behavior episodes in some children with autism.

Other studies [84,98] included the time and domain frequency features of PPG. PPG can also calculate several essential measures such as heart rate (HR), interbeat intervals (IBIs), heart rate variability (HRV), and heart rate reserve (HRR). HRV is influenced by activities such as exercise, eating, and sleeping, is related to emotional arousal, and decreases when a person is stressed [91]. Ref. [91] used discrete wavelet transform (DWT) for analyzing ECG, where there were extracted features such as db2, db4, and db8. In addition, they used the Pan–Tompkins algorithm to derive HRV and QRS amplitude.

Dutheil et al. [91] mention that SC-series and HR-series will detect different responses and may be complementary. SC is related to cognitive arousal, and SC is augmented with stress, and it tends to stay low or drop when a person is less aroused (disengaged, bored or calm) [92]. SC, also known as GSR or EDA, can be measured in a range from 0.01 μS to 100 μS (Siemens). EDA is closely linked to the sympathetic nervous system, which is responsible for the body’s arousal and stress responses. This non-invasive measurement of the electrical conductivity of the skin has the potential to provide valuable insights into an individual’s emotional and cognitive states.

A study [99] used GSR signal feature extraction such as mean, standard deviation and rate of change. Another study [87] found that GSR detection can be influenced by parameters such as the anatomical location of the measurement, the interelectrode distance, and the electrode material. The authors compared the signals measured using four wearable GSR sensors worn on the right and left wrists, the fingers of the left hand, and the right foot, and they found better results on the fingers. Pfeiffer et al. [87] used EDA to evaluate auditory hypersensitivity in children with ASD, where they used Empatica E4 collected EDA data and analyzed differences between SCL and SCR, which are associated with chronic and acute stress. In ref. [86], the EDA signals were evaluated and identified; in the emotional state anger, EDA increases and EDA peaks are big and few, while sadness EDA peaks are small and many.

Some of the extracted features were the mean of peak amplitudes, the median of peak amplitudes, the standard deviation of peak amplitudes, the root mean square of peak amplitudes, the maximum of peak amplitudes, and the minimum of peak amplitudes, among others. The features of a GSR consist of latency, peak, amplitude, rise time, and recovery time. Normally, GSR requires two electrodes applied to the skin, usually on emotionally sensitive locations such as the palm, fingers, and the sole of the foot.

One study used accelerometer (ACC) data [100], which are used as output for stress detection [101]. This sensor can help to identify the body movements. The accelerometer and gyroscope data can be used to detect moments of behavioral crises [80]. The authors of [83] investigated the comfort, robustness, and validity of a commercially available ambulatory cardiovascular monitor for measuring physiological stress in children with ASD. However, sessions, which were video recorded, found that rigorous child movement accounted for the issues with robustness in most cases. Therefore, authors proposed to record cardiovascular responses and accelerometry simultaneously to examine the impact of user movement on robustness metrics.

Some studies have used ST as an indicator of stress [18,86]. Some of the reviewed studies used the Empatica E4 watch to measure ST [102]. The assessment of ST has shown that under acute stress, the sympathetic nervous system triggers peripheral vasoconstriction, resulting in short-term changes in skin temperature. Normally, the range is from 32 to 35 °C [93]. ST has the potential to provide information about the intensity of the stress response. The temperature sensor in the E4 watch uses an optical infrared thermometer with a resolution of 0.02 °C.

Other studies used features such as NN20, NN50, pNN20, pNN50 [91], the standard deviation of all R-R intervals (SDNN), and the square root of the mean squared difference of successive R-R intervals (RMSSD). In addition, studies used the Pan–Tompkin’s algorithm, which is a widely used method in feature extraction for the automatic detection of QRS complexes in ECG signals, which are key for identifying heartbeats.

Figure 5 shows a summary of sensor locations used in the reviewed studies. Physiological stress produces an increase in the electrical activity of the sympathetic nervous system (SNS), causing changes in the physiological responses of the body, such as cardiac activity, respiration rate, electrodermal activity, and skin temperature, which are the responses most frequently evaluated in the studies found. However, other responses such as movement and EEG were also used.

A study used noise detection protocol such as an acceleration sensor to identify and exclude periods of excessive movement [57].

**Question** **3.**
*What techniques/processes are used for stress detection in individuals with ASD?*


The studies that we found relate stress to valence/activation detection. Valence refers to how pleasant or unpleasant an emotion is. High valence indicates positive emotions, while low valence indicates negative emotions. Arousal, on the other hand, refers to the level of arousal associated with an emotion. High arousal indicates intense emotions (such as anger or surprise), while low arousal indicates calm emotions.

For arousal detection, the GSR signal captures a numerical value that is sampled at a certain frequency, which can vary depending on the purpose. A study [91] proposed four phases to identify stressful situations such as (1) 24-h baseline pre-experiment (physical activity, sleep); (2) 2 h in a real-life situation; (3) 30 min in a quiet environment, interrupted by a few seconds of stressful sound; and finally, (4) an interview to record feelings about events triggering anxiety.

Another study [57] designed a set of tasks such as exercise, motor imitation, playing together, manipulating, reading, and vocal imitation. The tasks were conducted during the sessions, while ECG was measured using the developed device. Therefore, the authors evaluated acceptability and quality of the ECG signal. Additionally, the behaviors were recorded by video cameras. However, the experiment was applied with four participants with ASD, where three consented to wear the developed clothes.

One study [93] used cardiac activity for stress detection, where the researchers used features such as pNN20, NN20, NN50, mean RR, mean HR, and mean RR and applied machine learning classifiers such as SVM and LR, with LR models achieving 93% and SVM 87%. Tomczak et al. [18] proposed an algorithm for stress detection based on a heuristic rule-based system with the following main assumptions: (1) a decrease in resistance is observed at ∆t after a pulse increase; (2) a temperature decrease is observed at ∆t after a decrease in resistance; and (3) a temperature decrease is observed at 2∆t after a pulse increase, where ∆t is an experimentally evaluated constant. The algorithm considers measures such as pulse, skin resistance, and body temperature.

One study [86] designed an algorithm for emotion recognition using physiological signals such as EDA, HR, and ST. The authors analyzed changes (increase/decrease) for each emotional state. In addition, the EDA signal is analyzed in terms of size and number of peaks, i.e., anger EDA peaks are big and few, with HR slight decrease, EDA increase, and ST slight decrease. In ref. [103], forty children with ASD between 4 and 11 years and their primary caregivers participated using an EDA sensor with different tasks such as free play, clean-up, dyadic problem-solving, and frustration, where they analyzed the variability in EDA for each child within each task. EDA was recorded in μS at 8 Hz using Affective Q-sensors. The authors concluded that EDA is complex, because it requires when, how, and in what context the responding occurs.

Ref. [103] considered a set of rules for the quality of EDA data such as (1) EDA is out of range where they chose 0.05 μS as a minimum for SCR amplitude; (2) EDA changes too quickly; (3) EDA is affected by a range in temperature >40. Therefore, they analyzed the data segments, and they found both valid and invalid data. The authors concluded that they should consider EDA hardware, participant sample, and the relative importance of keeping vs. removing both valid and invalid data.

Ref. [86] used the Empatica E4 to classify emotions. Therefore, they began capturing physiological signals such as EDA, SKT, and HR. Then, each signal was analyzed comparing their changes (increase/decrease). To extract more knowledge from the signals, they analyzed the peaks; for example, in EDA, if a peak height is between 0.05 to 0.4 μS, it is classified as a small peak, and if a peak height is more than 0.85 μS, it is considered a big peak. Therefore, statistical analyses were categorized into three groups: (1) signal with over 10 peaks; (2) signals with more than 2 and fewer than 10 peaks, and (3) signals with exactly 1 peak. Finally, for the classification, they used a decision tree based on statistical information. For example, if there are more small peaks in the signal, the counter for the happiness, anger, and sadness is increased. The authors calculated the probability for each emotion showing certain characteristics and added different weights to the counters. Emotions such as anger and sadness demonstrated similar characteristics for small peaks, but the statistical analysis shows that in the presence of many small peaks, it is more likely that the test subject is sad.

In ref. [89], a personalized protocol was developed for every child with ASD to induce positive and negative valence. The experimental protocol for each child had three trials with ten minutes of break in-between each trial, which lasted for 30 to 40 min, where they used audio and visual stimulus (film songs, cartoons, rhymes, news, sports). ECG signals were captured at the sampling rate of 512 Hz. Next, they applied DWT to remove the corrupted data due to movements, breathing, and behavioral activities. Furthermore, the high-frequency noise signals were removed using an order low pass Butterworth filter with a cut of frequency of 20 Hz [92] or using an IIR notch filter at 50 Hz and an IIR low pass filter at 40 Hz [101].

Then, the final step of the pre-processing phase implements the Pan–Tompkins’s algorithm to detect the QRS. Finally, in ref. [89], the applied classifiers such as SVM, ensemble and KNN obtained a maximum accuracy of 83.8%, 87.4%, and 81%. In ref. [101], the authors proposed a method for ECG analysis where features such as heart rate, RRMSSD, and RSA were extracted, where the physiological events detected showed a higher HR, lower RMSSD, and lower RSA. In addition, lower HR values were more frequent during the child’s inactivity.

Ref. [104] evaluated different machine learning algorithms such as GBDT, KNN, RF, and SVM for the detection of stress and attention. The authors used data captured by an EEG headband on children with ASD.

**Question** **4.**
*What techniques are applied for multimodal physiological signal fusion?*


The analysis of physiological data consists of two steps: preprocessing and feature extraction. Multimodal stress detection involves the use of multiple types of data [18,91] (such as physiological, behavioral, and contextual data) to detect stress levels. For individuals with autism, multimodal stress detection is particularly important due to the unique ways that they may express or experience stress. Various techniques and sensors can be employed to capture physiological responses, such as changes in HRV, skin conductance, skin temperature, and changes in breathing patterns, among others. Multimodal data fusion refers to the process of integrating information from multiple sensors or data sources to obtain a more accurate, reliable, and comprehensive understanding of a person’s physiological state.

When applied to stress detection, for example, multimodal data fusion can combine signals such as heart rate, electrodermal activity, respiration rate, and skin temperature to detect stress more effectively than any single signal could. Some types of data fusion in multimodal physiological signal processing are (1) sensor-level fusion; (2) feature-level fusion; (3) decision-level fusion; and (4) hybrid fusion. However, little is mentioned about it. In most studies, the signals are analyzed separately to detect patterns. Tomczak et al. [18] used a rule-based system combining information from different sensors such as heart rate, skin resistance, and body temperature. Ref. [92] combined physiological characteristics BVP and EDA. Moreover, in ref. [105], the features extracted for each sensor, in total 8 between GSR and PPG, are classified using an SVM classifier to identify emotions in ASD.

The studies reviewed combined two physiological signals such as EDA and ECG, and very few used three sensors.

**Table 1 sensors-24-08137-t001:** Summary of published research on wearable solutions for stress monitoring in individuals with ASD.

**Study/Year**	**Device Type**	**Sensors**	**Signals/Parameters**	**Outcomes**	**Purpose**	**Population**
[80]/2014	Affectiva Q Sensor; MindWave Mobile; Zephyr BioHarness	ACC; ST; GSR; EEG power spectrums; ECG	Arousal, alpha and beta waves, HRV.	-	App mobile for Monitoring emotions such as happiness, sadness, fear, disgust, surprise, and anger.	-
[105]/2014	Mynplay Brainband	EEG; GSR; HR	EEG waves	-	Analyzing mental states	-
[91]/2015	Zephyr BioHarness BT; two-lead cardiscope One.	ECG; SC; RT	RR-Intervals, HF and LF energies, time series SC and RT	Resulting stress induces a bilateral variation in HR of 60.8 ± 1.3 bpm and 5.2 ± 0.2, and SC ranges from 3.65 ± 15 μS to 3.31 ± 0.13 μS	A protocol was conducted for monitoring physiological responses for individuals with ASD	Thirty participants with ASD between 10 and 25 years
[57]/2016	Shirt	ECG; ACC	R-R Intervals; Pan–Tompkins algorithm	Three participants accepted the wearable. The duration of the sessions was 60 min; different tasks were applied; tasks with high SD were reading, card selection, and motor imitation.	Design a smart wearable for monitoring	Four male children with ASD between 5 and 8 years
[92]/2016	Wristband	GSR; PPG	SD-GSR; mean-GSR; HRV (LF, HF), IBI	SVM is employed for detection of emotions, with accuracy of 90%, with specificity of 100%, and sensitivity of 80%.	Monitoring emotions such as happy, neutral	Ten children with ASD
[86]/2017	ECG chest belt	EEG; ECG	RMSSD; RSA; HR;	The children did not show sensory-motor and/or behavioral issues in wearing the devices and completing all the tasks. ECG patterns showed similar changes in five patients in a socio-cognitive task.	Propose a method for ECG analysis of data collected.	Five patients with ASD between 6 and 8 years; one session a week for six months
**Study/Year**	**Device Type**	**Sensors**	**Signals/Parameters**	**Results**	**Purpose**	**Population**
[86]/2017	Empatica E4	EDA; ST; HR	RMSSD; SD; SDNN	Happiness: HR (slight increase); EDA increase; EDA peaks (small and few); SKT (slight decrease). Sadness HR (decrease); EDA (Increase), EDA peaks (small and many) and SKT slight decrease)	Classifies emotions such as anger, happiness, pain and sadness	Ten subjects between 20 to 25 years.
[88]/2017	LG Watch Urbane	PPG, ACC; barometer	-	User B experienced strong anxiety moments whose manifestation was not easily visible. User A manifested several signals of fear and tension.	A system for emotional self-regulation	Two individuals with ASD ages 10 years were tested 4 h a day over nine days
[87]/2018	-	ACC; ECG; blood glucose	-	-	Designing a wireless body area network	**-**
[106]/2018	-	EDA	SCL; SCR	Some data were missing due to equipment malfunction and/or removal of sensors. EDA were significantly related to externalizing behavior scores.	Testing electrodermal activity through 2 tasks	Forty children with ASD between ages of 4 and 11 years.
[82]/2018	Thoracic belt	Piezoelectric sensors, ECG	SD; mean	Three game activities were evaluated, where in the first game activity, the HR is higher than mean frequency of 96 bpm. The last game activity, in which the child plays with a balloon inflated by the operator, is characterized by a great variability in HR, i.e., 102.9 bpm.	Testing the device during the designed activities	Five children with ASD between 2 and 5 years
[103]/2018	Affective Q Sensors	EDA	SCL; SCR	Four simple rules, such as (1) EDA is out of range (not within 0.05–60 μS); (2) EDA changes too quickly; (3) Temperature is out of range (not within 30–40 °C); (4) EDA data are surrounding (within 5 s)	Testing data quality of EDA considering 4 rules.	Fifteen males, 5 females, ages range 5–13 years for 8 weeks
[99]/2018	-	GSR		The lowest value of EDA was observed when the sensors were placed on the middle and index fingers. The data collected on the wrist and forearm were comparable.	Validation in four parts of the body such as (1) on the arm; (2) on the leg; (3) on the leg, 1—index and middle fingers, 2—wrist, 3—forearm, and 4—ankle.	Sixteen subjects between 15 and 50 years (8 males and 8 females)
[97]/2019	Biopac AcqKnowledge versión 4.4.2	ECG	RMSSD; QRS; BPM; HRV	BPM before challenging behavior was higher (SD = 16.10). HR increase was significantly associated with challenging behavior.	Testing HR activity in tasks designed to induce low-level stress.	Forty-one children with ASD between 2–4 years (32 males, 9 females) sessions 1 to 1.5 h.
[93]/2019	Chest strap ECG sensor	ECG	Max, Min, mean, SD features of HRV; RMSSD, NN; NN20; NN50; pNN20, pNN50	Applied algorithms such as LR and SVM with 84% and 91% accuracy, respectively.	Stress detection using two classes: rest—stress, using two tasks designed to mimic stressful scenarios, such as transparent box and tangram/tangoes.	Twenty-two participants with ASD
[87]/2019	Empatica E4	EDA	Arousal	Two different types of noise-attenuating headphones were designed. EDA was recorded continuously with a minimum of 20 min. Decibel levels increased; SCL increased during the stages without intervention.	Evaluate the proof of concept of an intervention to decrease sympathetic activation using EDA	Six participants with ASD between 8–16 years.
[18]/2020	Wristband	PPG; GSR; ST	HRV; BPM; GSR; ST	Not all participants understood the purpose of wearing the device. Each decision to classify a certain situation as stressful was based on symptoms such as behavior, gesture, interaction, voice, and facial features.	Device for monitoring stress levels	Twenty participants between 5–24 years (19 males; 1 female), wearing time: 5 h.
[89]/2021	Shimmer ECG	ECG	HRV; QRS; Pan–Tompkin’s algorithm; mean; median; entropy; kurtosis	HRV and QRS amplitude were classified using KNN, SVM, and ensemble classifier, obtaining accuracy of 81%, 87.4%, and 83.8%.	Protocol to induce positive and negative valence and ECG	Six children between 5 to 11 years
[104]/2021	EEG headband	EEG	EEG; GSR; ACC; ST; HR	GBDT, RF obtained the best prediction accuracies of 86.67% and 99.05%.	Evaluate machine learning models such as GBDT and RF	Thirty-five participants with ASD
[107]/2021	ECG Comftech CozyBaby	ECG	RR; SD1; SD2	We analyzed a total of 26 therapeutic sessions.	Variability in the therapist’s heart rate and conversational turn-taking during online sessions.	Sixteen participants between 6 to 18 years with ASD level 1.
[108]/2021	Bracelet	EDA; PPG	RR	-	Design a smart bracelet	-
[83]/2022	Polar, Mio Muse and PulseOn	ECG;	HRV	Three wearables (Polar H7, Mio Fuse, and PulseOn) were within a priori sampling fidelity.	Evaluating commercial wearables for elements such as stressor task; comfort	Thirty-two children with ASD between 8–12 years
[109]/2022	_	ECG; EDA	SCL; SCR; HRV; RMSSD	Affective sliders, and state trait anxiety scale questionnaires were used as self-reports.	Include physiological data to evaluate social interaction behaviors in children with ASD	Seventy-two children between 8–12 years. (female = 12, male = 60)
[22]/2022	Empatica E4	EDA; PPG; ST	LF; HF	Each interaction session was analyzed in detail in 2–3 min intervals and validated with emotional labels.	Testing physiological-signal-based stress detection to be used in social interaction	Twenty-nine children between 2 to 12 years; 2–11 sessions
[110]/2022	-	PPG	RMSSD	Participant age was significantly negatively correlated with all HRV variables, namely, high-frequency HRV. Some difficulties related with device and problems with home testing of HRV.	Exploring HRV biofeedback	Twenty participants with ASD between 13–24 years
[85]/2022	Hexoskin	ECG	QRS	Ten participants with ASD who have aggressive or disruptive tendencies were monitored for 7 days. In addition, LSTM was applied.	To assess the differences in physiological reaction to stressful stimuli	Twenty participants with ASD between 20–40 years for 7 days
[111]/2023	Empatica E4	PPG, EDA	Average, SD of HR; SCL; SCR (number of peaks, amplitude, rise time)	A total of 207.6 and 203.5 h of physiological data that reported stressful events, where PPG and EDA were evaluated.	To understand the stress through physiological signals	Eight children between 5–12 years for 2 days
[98]/2023	Polar OH1 or Verity sense	PPG	HRV	If HR increases by more than 2 SD above the average HR, it sends a signal to alert the caregiver. HR response to events is not a specific measure of pain per se.	Heart rate monitoring to detect acute pain in non-verbal patients	Thirty-eight participants with ASD for 2 weeks
[84]/2023	Samsung Galaxy 3	PPG	HRV, DWT	The average HR of all participants is 96.8 BPM, and max HR is 124 BPM.	Analyze affective states	Nine children between 8–11 years (6 male and 3 female)
[112]/2024	Ballistocardiography	RR; HR	Filters, peak detection.	Results show reasonable frequency estimation performance both for the respiration rate and heart rate.	Emotion detection for individuals with ASD who face difficulties in communicating their emotional stress and discomfort during medical or dental visits.	-
[94]/2024	Sewn-in pockets (BioNomadix ECG) and ACC module; Polar H7; Mio Fuse	PPG; ACC; ECG	QRS; SD, mean; BPM	For all children, mean heart rate and peak heart rate were extracted from the SW for the low-level stress task and resting state periods.	Testing low-level stress through physiological signals	Forty-one children with ASD between 2–4 and 8–12 years

## 6. Discussion

The studies reviewed have explored biomedical sensors as potential indicators of stress such as heart rate, skin conductance, and temperature. However, there are several complications and challenges that must be considered. In the studies reviewed, these solutions have not been tested in long-term real-world use. In addition, there are ethical and privacy concerns. Finally, the studies reviewed were more about analyzing the patterns of each sensor on children’s behavior for a set of tasks to be performed, so the design of autism-focused technologies was not evaluated, especially the impact of potential sensory impairments, multi-sensory integration, and attention challenges.

Stress monitoring involves two processes: data acquisition and classification of the acquired data. Data acquisition includes biosignal acquisition and signal preprocessing [113]. The biosignals mostly used in the reviewed literature to monitor stress are GSR, PPG, TMP, and ACC. Signal preprocessing includes the amplification, filtering, averaging, and extraction of relevant features to be used as training data for the classifier. Algorithms used for the classification included k-NN, SVM, and LR.

Tomczak et al. [18] mention that commercially available applications for stress monitoring are limited, are not easy to use, and generally require the user to remain seated for a certain period of time. These monitoring apps expect an individual with ASD to be able to monitor themselves rather than a supervisor being able to monitor the signals if that individual is a minor [27]. In addition, many of these devices were created for fitness users. Very few were created for therapy interventions; one example is the Empatica E4 wristband, which is a portable and wireless device that collects physiological signals such as ST, EDA, PPG, and ACC in real-time. However, the E4 Empatica is not affordable as access to the data requires payment. This device can be used to record physiological signals in two ways: (1) in real time and (2) with the user storing the data locally on the device. However, the application does not offer data visualization associated with stress, which could be useful for monitoring stress through a mobile application. Not only should wearables for autism capture the physiological signals of the person with ASD but they should also be notified in real time through a smartphone application, and the device should also have the option for another person, such as a caregiver, to monitor these signals.

The reviewed studies (Table 1) involve various devices such as wristbands, chest straps, and shirts, among others. Each device has a certain size, material, and sensor number. Only one study evaluates the acceptability [57], but the designed garment may change depending on the child’s physical build. Most of the studies used commercial devices. Very few proposed a new one such as [10], which designed a wearable device type wristband containing electronic components and electrodes, capturing physiological signals such as heart rate, skin resistance, and body temperature, and proposed an algorithm for heuristic rule-based stress detection.

Motti and Caine [114] defined twenty principles related to human factors that should be considered during the design phase of wearables, namely, esthetics, affordance, comfort, contextual awareness, customization, ease of use, ergonomics, fashion, intuitiveness, obtrusiveness, overload, privacy, reliability, resistance, responsiveness, satisfaction, simplicity, subtlety, user friendliness, and wearability. Francés-Morcillo et al. [115] proposed 10 requirements that must be considered when designing a wearable: comfort, safety, durability, usability, reliability, esthetics, engagement, privacy, functionality, and satisfaction. The design of wearables involves human factors. One study [116] mentioned that the texture of the wearable could be presumably used to alert the user to the fact that stress has been detected; for example, other devices may use visualization to present visual information. Therefore, wearable devices could be designed to assist with sensory integration emotional regulation; these technologies can potentially offer more effective and personalized support [117]. The diverse needs and heterogeneity within ASD necessitate a personalized, user-centered approach to the design and evaluation of wearable technologies [117]. Participatory design methods that engage individuals with ASD and their support networks can help identity the specific features, from factors, and interaction modalities that are more acceptable and beneficial for this population. Furthermore, these devices can be designed to be unobtrusive and comfortable, reducing the risk of sensory overwhelm or discomfort for the user.

One of the challenges is how to design wearables for which the information is readily available to people with ASD outside the clinical setting that are well tolerated by them and that have regulatory approval. Many of the devices examined in the reviewed literature are not specifically tailored to the unique needs of people with ASD. These individuals, however, have sensory sensitiveness, so the device must be user-friendly and comfortable when worn during daily activities in the sense of not causing any discomfort to the user. That study also evaluated the reactions to wearing the wristband, obtaining very positive results. However, it did not describe the level of autism of the individuals who were assessed. Umair et al. [90] evaluated the user acceptance of the six most common commercial wearables for monitoring (Bitalino, Bodyguard, Polar H10, Polar OH1, Samsung Gear 2, and Empatica E4). The authors conducted interviews to assess wearability, comfort, esthetics, social acceptance, and the long-term use of each device. Some participants felt the Empatica E4 device to be a little heavier and more uncomfortable than other wrist-worn devices because of its electrodes constantly pressing against the skin. However, Empatica E4 has more sensors compared with other devices. There may be advantages to using more sensors, which can provide valuable information to assess stress levels. However, an optimal combination is required to achieve higher overall accuracy, with most research focusing on a single type of sensor. Few studies have focused on the use of multiple sensors for stress monitoring. Multimodal techniques integrate data from multiple sensors or modalities and behavior patterns associated with ASD [118].The reviewed studies did not evaluate the effect of combining several different signals and the implications of data acquisition procedures and details.

According to Koumpouros and Kafazis [116], the most important factors for the design of wearables are durability and comfort, because these are crucial for long-term monitoring. They also reported a preference for devices that are made of flexible materials and are soft and easy to wear and remove when needed. The authors also mentioned that when devices are designed for individuals with ASD, they should take into account the cost of devices when designing them to facilitate their use by people with ASD and their caregivers.

Wearable technology has potential as a tool for monitoring stress [18,57] and anxiety [65] detection, and self-regulation [88] in ASD. However, it requires evaluation of the long-term effectiveness of these technologies in real-world settings. However, these devices face a common challenge in terms of signal quality [119]. In addition, there are sensors such as the gyroscope or accelerometer that have specific requirements in terms of body position and posture to provide accurate and reliable measurements. The authors of another study examined the effects of wearables for anxiety detection, but the study had limitations such as a small sample size, imbalanced group matching for IQ and sex, and controlled laboratory settings.

The studies reviewed used one sensor or two, but using multiple sensors is required to apply data fusion techniques, where these data streams can be integrated and analyzed using advanced machine learning techniques to develop robust stress detection models. However, a critical step in the development of machine-learning-based stress detection systems is the acquisition of high-quality training data. Physiological data, such as heart rate, skin conductance, and respiratory patterns, can be collected from wearable sensors worn by participants during various stress-inducing activities or situations [96]. However, stress-inducing activities for participants with ASD must be designed with the help of a therapist and should be supervised with parental assistance. In addition, sensory-related behaviors, such as hypersensitivity to certain stimuli, can significantly impact the child’s and family’s ability to engage in routine activities, both inside and outside the home. One study [94] used the transparent box task from the preschool and middle childhood versions of the laboratory temperament assessment battery as a low-level stress task to induce frustrations and mild distress in children with ASD. Therefore, this makes the studies much smaller, because it requires designing the activities well and having a therapist and parental support. In addition, measurements can take periods of time and even days. Ref. [91] mentions that time-series analysis has a long history in econometrics but in health, it is shorter; it requires analysis of a wide variation of physiological signals with different types of activities in real environments.

Therefore, to evaluate the performance of a portable detection system, the experiment should be performed in a controlled laboratory. In addition, the experiment should include structured periods of relaxation and low-level stress to investigate the correlation between self-reported stress and physiological biomarkers. In addition, it should be considered that there may be corrupted data due to motion artifacts. Ref. [101] presented some limitations with the study design such as (1) HR responses to events are not a specific measure of pain per se and thus must be interpreted in the situational context to be meaningful; and (2) stressors can change for each participant. That involves conducting larger studies to establish normative values for HRV in persons with ASD under a variety of conditions. Several previous studies assessing the physiological response of persons with ASD have indicated that this population may have different responses to stress compared with their neurotypical peers.

When people with ASD experience stress, it is highly amplified, so it can be useful to identify stress-induced physiological state changes. To recognize stress, it is necessary to train models on physiological signal data associated with stress crises for individuals with ASD. Many of the studies found in this review used machine learning techniques such as SVM, LR, RF, KNN, and CNN for stress detection. For the training and validation of the stress detection model, they used datasets such as WESAD, PhysioNet, AffectiveROAD, and DEAP. In addition, the datasets were centered on one type of device, and this may affect the detection model if the device is changed. Furthermore, the datasets were focused on individuals without ASD, and a future challenge will be to design models trained on datasets that include people with ASD; currently, no such datasets are available. However, there are new efforts in this area, with Aktives for children with different special needs [120] and autism. In addition, to build these datasets, a protocol needs to be designed to induce stress, but in autism, this is a risk.

Datasets are built applying methods for labeling and categorizing levels of stress. Some methods were employed such as SCWT, Stroop-CW, trier social stress test (TSST), sing-a-song stress test (SSST), and visual analogue scales can be used for inducing stress. However, it requires the importance of understanding the impact of stress and unpredictability on the cognitive and behavioral profiles of individuals with ASD, as well as the potential for using methods that induce stress as a means of building comprehensive datasets for the development of more effective interventions.

## 7. Conclusions

Studies on wearable solutions for individuals with ASD are limited. Wearable technology can provide an alternative for monitoring stress or therapy interventions for parents and caregivers. We also observed that in the selected publications, the physiological signals mostly used were cardiac activity, EDA, and ST. According to the studies reviewed, SCR and HRV are the most studied of all physiological parameters. Measurement accuracy is a challenge that can significantly affect stress detection. In addition, some of the reviewed studies used accelerometers, showing that they can be relevant to stress detection.

When reviewing the literature, it was found that machine learning techniques applied to stress detection used commercially available wearable devices or datasets such as WESAD, PhysioNet, and AffectiveROAD, among which the most used was the WESAD dataset. However, there is a lack of large, diverse public datasets.

The studies reviewed mentioned how to capture data, data quality, technology acceptance, and use but not how wearable technology could be designed for individuals with ASD. Therefore, it should be noted that none of the selected studies discussed UX factors or key problems with noise in physiological signals that could be occasioned, such as (1) Individuals with ASD often engage in repetitive movements (stimming) or sudden gestures, which can disrupt signal accuracy; (2) Poor attachment to wearable devices can cause inconsistent readings. For example, loose wristbands may result in gaps between the sensor and skin; (3) simultaneous measurements from multiple sensors can introduce overlapping signals or interference, particularly in compact wearable designs.

In addition, the most used sensors were the Empatica E4 and Shimmer. Empatica is the only device that has been clinically validated with children with ASD. These commercial devices can vary in form factors, materials used, parameters used, compatibility with wireless connectivity, and remote monitoring. For example, connectivity is via Bluetooth, meaning the person wearing the wearable must have a mobile device and an app, not that another person not present can monitor the signals.

On the other hand, the reviewed studies do not mention the problems that can arise when working with different types of sources, which is a topic related to data fusion. In addition, an important requirement in ML techniques following a supervised approach is to have valid labeled data. Some methods employed for labeling levels of stress were found, such as (1) specific stress/no-stress periods during sessions where users are watching images or videos, i.e., inducing positive emotions, and then applying stress tasks such as SCWT, Stroop-CW, trier social stress test (TSST), sing-a-song stress test (SSST), and visual analogue scales; and (2) self-reporting via questionnaires such as temperament assessment battery, NASA task load index (NASA-TLX), depression anxiety stress scale (DASS), and perceived stress scale (PSS).

On the other hand, stress levels were evaluated in terms of two classes (rest and stress), three classes (high, medium, and low; neutral, stressed, and amused), or four classes (low, medium–low, medium–high, and high). Finally, a common limitation in all studies reviewed was the number of subjects and devices available to capture physiological data due to financial, human, and time constraints among the academic research groups.

## Figures and Tables

**Figure 1 sensors-24-08137-f001:**
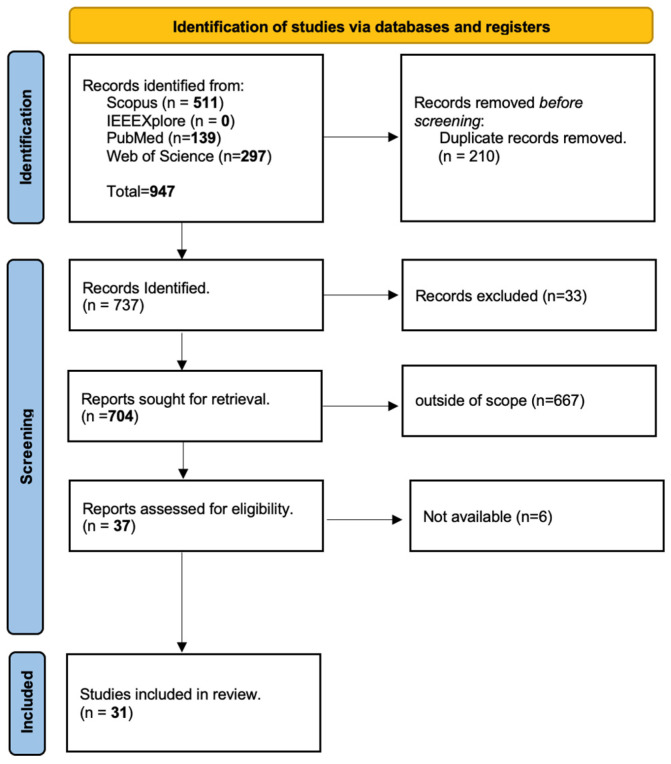
Flow of information of the systematic review process applying PRISMA.

**Figure 2 sensors-24-08137-f002:**
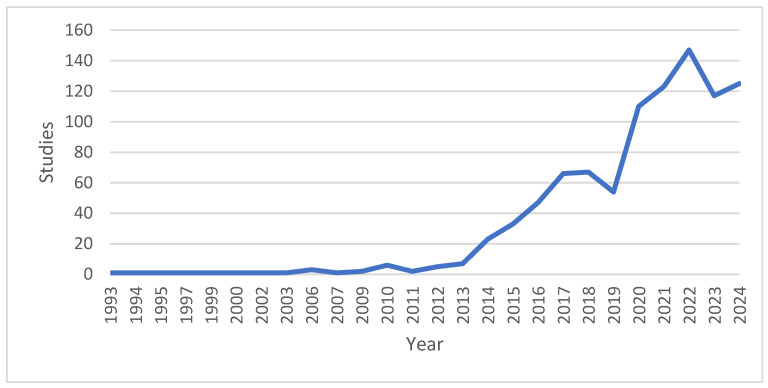
Studies published by year.

**Figure 3 sensors-24-08137-f003:**
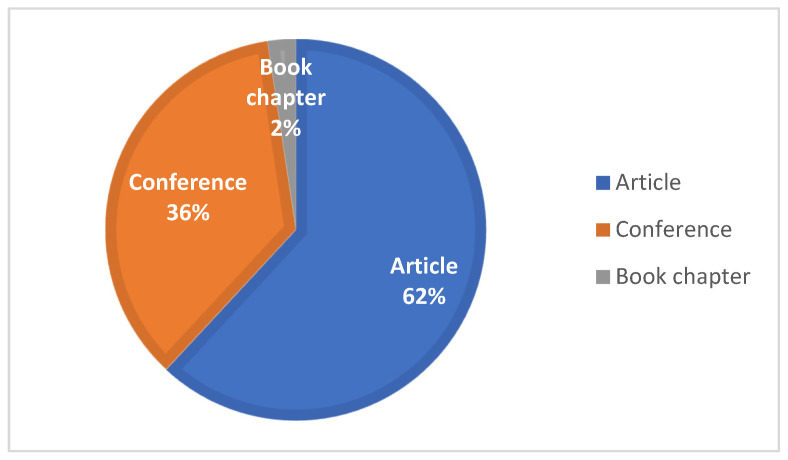
Document type.

**Figure 4 sensors-24-08137-f004:**
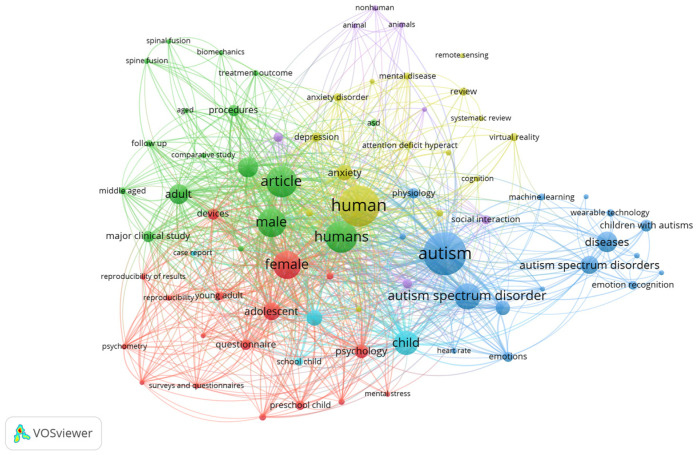
Studies by topic.

**Figure 5 sensors-24-08137-f005:**
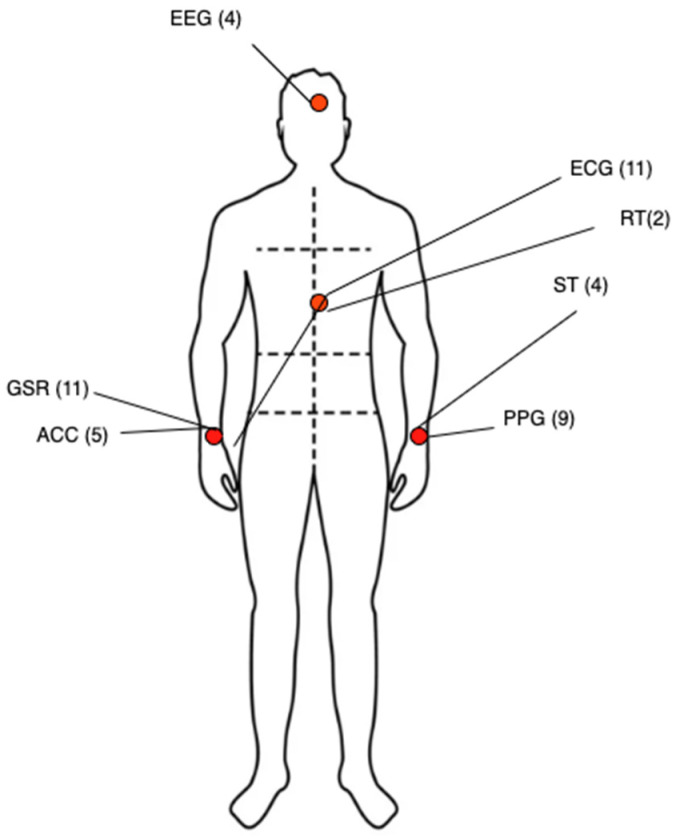
Body locations of physiological sensors in the reviewed studies.

## Data Availability

Not applicable.

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
