# Peer review of "Wearable Solutions Using Physiological Signals for Stress Monitoring on Individuals with Autism Spectrum Disorder (ASD): A Systematic Literature Review"

_sensors, 2024, doi:10.3390/s24248137_

Round 1

Reviewer 1 Report

Comments and Suggestions for Authors

Dear Authors,

The submitted paper presents a systematic review to identify wearable technology that detects stress in people with ASD. As the objectives presented in Section 3, the authors would like to find responses to the following questions: 

1- What kind of wearable solutions are most acceptable for people with ASD?

2- What features and sensors are used for stress detection?

3- What techniques/processes are used for stress detection in individuals with ASD?

4- What techniques are applied for multimodal physiological signal fusion?

However, the manuscript's text presents several concerns and should be enhanced by the authors. In particular, the authors should include some information about the disorder ASD and the population considered. In several moments, they mention ASD individuals, but in the citation, the work is a study about children or adolescents. They should also include information about stress and the difficulty of creating appropriate mechanisms to detect or identify it. Some affirmations in the manuscript simplify the problem of stress identification independently of the people (with ASD or not). This area has advanced in recent years, but the authors do not mention several complications considered in stress identification with physiological parameters. 

The manuscript lacks clarity and coherence. The introduction should have delineated the challenges of identifying stress in typical individuals and underscored the significance of identifying stress in individuals with autism (ASD). It should also have highlighted the important points to stress identification for this population and the difficulties related to the disease. This would enhance the research's relevance and impact. Regrettably, the introduction contains factual inaccuracies and should be revised accordingly. I have emphasised some critical points to review.

Title: The acronym for Autism Spectrum Disorder (ASD) should be used only once, appearing in the title, abstract, and introduction.

Keywords: It is strongly recommended to consider incorporating different terms in the title. Utilizing mesh terms, which can be identified through databases such as PubMed, would be beneficial.

Text:

Line 46: The statement "Stress in individuals with ASD is very common [2], with prevalence rates varying between 11% and 84% [3]" references the works of "F. Salazar et al." and "S. W. White et al." However, the prevalence rates mentioned are related to parental distress in the cited studies, not stress in individuals with ASD. Furthermore, both studies focused on preschool and elementary school-aged children. A thorough review of lines 46 to 53 is strongly recommended. Additionally, reference number 6 is incomplete and requires attention.

Line 69: The assertion "Studies have used physiological measures to monitor stress in people with ASD [9]" cites the work of "J. A. Healey and R. W. Picard," which is focused on stress detection in drivers, not individuals with ASD. Consequently, the progress in monitoring stress with biomarkers over recent years should be explained in this context.

Reviewing lines 78 to 106 is required.

Line 85: The claim that "The E4 wristband was designed for autism and includes photoplethysmography for heart activity, a three-axis accelerometer for movement, and an optical infrared thermometer for detecting skin temperature [11]" is inaccurate. The E4 wristband, developed by Empatica, is not exclusively designed for autism and has been utilized for biomarker monitoring in various studies. It is advised to reference a scientific paper as opposed to a product webpage in this context. Ongoing research in stress identification is essential due to the individualized nature of interpreting biomarkers and physiological parameters in people. The psychology community remains skeptical about standardizing stress identification, particularly through the use of parameters and AI resources.

Line 110: "Emotional states, such as stress, can be monitored via physiological responses [18]." This is not true. Stress is not an emotional state; it is the response of our organism to stressor agents, and the response is associated with an emotion. Physiological parameters can not identify an emotion. Good stress and bad stress generate similar physiological reactions. The reference number 18, cited by the authors, has the following information and sounds contradictory to what they wrote in the manuscript: " Apps which are evidence-based are capable of supplementing medical care, but not at specifically quantifying stress levels (Coulon et al., 2016). To quantify stress more accurately, researchers are developing devices that can give relevant, concrete data through the detection of specific stress biomarkers for stress monitoring. The recent developments in novel, non-invasive, wearable (or portable) sweat sensors will be discussed in further detail in the next section."

I sincerely suggest that the authors review Section 2 carefully.

Line 139-140: "Individuals with ASD, social exchanges and interactions may be difficult, and these social experiences often induce significant physiological stress [22]."  The reference cited is an essential study about adolescents: "Developmental effects in physiological stress in early adolescents with and without autism spectrum disorder". The affirmation regarding "Individuals with ASD..." ought to employ studies of relatives with adults or include other studies with this affirmation as citations.

Line: 186: "EDA, also called galvanic skin response (GSR) measures the conductivity of the skin and can be a useful indicator of stress [49]." I suggest including the Electrodermal Activity or EDA...

Methodology:

Additionally, the authors should refine the objectives and research questions in Section 3, as they currently appear overly general for a systematic review.

It is crucial to note that while the methodology is based on an adaptation of PRISMA, the approach employed for paper selection deviates from PRISMA's standard guidelines.

Also, the methodology should enclose a research strategy (PICO, PCC, SPIDER, ...), with a specific focus on including individuals across all age groups diagnosed with autism spectrum disorder, including children and adolescents. I didn't understand the following information from the abstract: "Our review found 31 articles; not all studies considered individuals with ASD, and some were beyond the scope of this review."

A systematic review is generally directed to a specific population, and the results compare different types of interventions and treatments. Other kinds of reviews should be included for various populations. I suggest that the authors review the methodology and the name systematic review used for this study.

Furthermore, the author's selection of a specific time frame, limited database investigation, and lack of mention of study risk of bias, reviewer members, and other pertinent information, such as the inclusion of grey literature, requires attention.

Tables: The acronym table should be included in the final section of the Abbreviation Section. Furthermore, the excessive focus on the string employed in presenting four tables is unwarranted. You should provide only one more organized table.

References: It is highly recommended to reevaluate the references to address excessive citations thoroughly. Review the presence of web pages and inconsistencies in the provided data; data are missing or undesirable, as the product page mentioned earlier. Additionally, it is recommended that the cited papers be assessed and integrated with more recent sources that specifically focus on stress identification.

The entire manuscript requires a thorough revision to identify all instances of incorrectly cited papers. It also needs to improve information about stress identification and physiological parameters in ASD people, which is widespread in Section 2. This theme is essential, and the study results may contribute to accelerating the influence of biomarkers and physiological parameters on stress identification. I want to insist on a deep review of the paper content submitted because stress identification is mainly essential to help ASD people and other diseases that require attention and healthcare. 

Hopefully, this feedback will help you improve your work. Demonstrating the importance of the sensors' monitoring to stress identification will help enhance their usage in several fields, mainly among health workers.

Regards,

Comments on the Quality of English Language

Dear Authors, I also recommend a good review of the English language. Several points in Abstract, Section 1, Section 2, and Section 3 need to be improved.

Reviewer 2 Report

Comments and Suggestions for Authors

This review focuses on the use of wearable devices to measure physiological stress in individuals with autism spectrum disorder. This paper structure is well-organized but the contribution to the study of Autism Spectrum Disorder is limited by the broad results and discussion.  The research questions have not been fully addressed, as the paper does not clearly highlight the specific considerations or adaptations required for wearable solutions tailored to individuals with ASD.

My suggestion is to do more research on the needs and requirements of sensors for populations with autism spectrum disorder, as people with ASD may be more sensitive to wearable devices. For example, in the 31 selected papers, the wearable sensors guide how to guide the ASD group to use sensors for high quality data, and how to identify the noise in the data due to sensory integration abnormalities.

Section 1-Introduction: well-organized and clear.

Lines 74-76, please explain why and how the wearable device can be used to monitor the stress levels of the ASD population in everyday life. What are the advantages and disadvantages of using the wearable device compared to other methods?

Section 2-4. The aims and methodology are well aligned with the research framework. However, please review the paper selection, did you exclude papers that did not focus on the ASD population?

Section 5. Results:

For question 1:

This paper aims to present different wearable solutions from 31 papers that are most acceptable to people with ASD.  However, there are only two papers that focus on people diagnosed with ASD. "Other studies focused on people without ASD".  Please clarify this point and explain how you have answered what is most acceptable for people with ASD from only two papers focusing on people with ASD?

For question 2:

The authors summarized the commonly used features and indexes for stress detection from 31 papers. However, as they previously mentioned, there is only a small number of studies focusing on individuals with ASD, making it unclear whether these stress detection methods are suitable for people with ASD, such as accuracy, measurement range, and usability.

For question 3-4:

The authors list the techniques and processes used for stress detection, but it looks general, as authors do not explain why these techniques and processes are specifically applicable to individuals with ASD. How do these methods differ from those used for non-ASD populations. Additionally, they did not highlight the unique challenges of measuring stress in individuals with ASD, such as potential delays in response or  data cleaning due to sensory sensitivities. Rather than listing the papers one by one, the review would benefit from focusing on some specific variables, features, and techniques that are particularly useful for stress detection in individuals with ASD.

6. Discussion

"The reviewed studies (Table 6) include different devices such as wristbands, chest straps, and shirts, among others." Did these papers mention how people with ASD accepted these devices differently? What is the most acceptable solution? How do these sensors work differently over different measurement periods (days, weeks or months)?

“In this review, studies using wearable device data as an intervention tool for individuals with ASD are limited.” What might be the reasons for this? And how can this be changed?

Round 2

Reviewer 1 Report

Comments and Suggestions for Authors

Dear Authors,

After thoroughly reviewing the reviewer’s response, I have analyzed the updated version of the document along with the authors' responses. I noticed that some recommendations were only partially addressed, while the most critical ones were overlooked. It is crucial to pay particular attention to Comment 4, as it has not been satisfactorily addressed.

I will elaborate on one of the points again and emphasize the necessity of revisiting all considerations from Comment 4 of the initial evaluation. Specifically, this issue compromises the overall quality of the submitted paper. The authors must conduct a comprehensive examination of all aspects related to the disease and its associated stress conditions. Following this, a thorough review of citations and assertions lacking strong theoretical foundations should be undertaken.

Please take note of all points, and I ask that you respond adequately, including all points of Comment 4 from the initial review analysis. The author's response was frustrating: "Response 4: I was updated."

The manuscript text should be read carefully:

"Stress in individuals with ASD is very common [2], with prevalence rates varying between 11% and 84% [3]. Stress can impact the physical and mental health of a person with ASD [4]. People with ASD may also be at a high risk of experiencing very stressful and traumatic events, which can negatively affect their mental health [4]. According to the DSM-5, approximately 70% of people with ASD have a comorbid mental health disorder, and up to 40% have two or more. Typically, people with ASD encounter problems related to sensory processing [5] and can thus experience sensory overload, where one or more senses react to stimuli, triggering elevated stress levels."

My point is as follows:

The first statement, "Stress in individuals is very common [2]," cites the reference: [2] F. Salazar et al., ‘Co-occurring Psychiatric Disorders in Preschool and Elementary School-Aged Children with Autism Spectrum Disorder’, J. Autism Dev. Disord., vol. 45, no. 8, pp. 2283–2294, Aug. 2015, doi: 10.1007/s10803-015-2361-5.

By citing a study focused on school-aged children, the authors have generalized their findings to individuals rather than limiting them to children or adolescents.

The most concerning part of the highlighted paragraph is when the authors state that the prevalence rates vary "between 11% and 84% [3]," citing the paper:

[3] S. W. White, D. Oswald, T. Ollendick, and L. Scahill, ‘Anxiety in children and adolescents with autism spectrum disorders,’ Clin. Psychol. Rev., vol. 29, no. 3, pp. 216–229, Apr. 2009, doi: 10.1016/j.cpr.2009.01.003.

As part of my analysis, I need to reference the cited paper:

"Methods:

This review was based on a systematic search of published articles available through August of 2008. The Psych-Info and Medline online databases were searched concurrently for entries containing any combination of the following terms: (1) autism, asperger(s), or pervasive developmental disorder and (2) anxiety or anxious. Abstracts of identified articles were then screened based on the following inclusion criteria: (a) the target population included school-age children or adolescents (between 6 and...).

Prevalence:

Large-scale epidemiological studies have not been conducted on the prevalence of co-occurring anxiety disorders in ASD. However, studies reviewed here indicate that between 11% and 84% of children with ASD experience some degree of impairing anxiety (Table 1). In the only studies examining diagnosed anxiety disorders in ASD, de Bruin, Ferdinand, Meester, de Nijs, and Verheij (2006) found that slightly more than 55% of the sample met the criteria for at least one anxiety disorder."

The manuscript presented by the authors lacks confidence. How can they assert that stress is prevalent in individuals when this data comes from a study focused on a different population? Furthermore, they are discussing children rather than individuals in general.

Regarding verification of this new manuscript version, the disorganization of Table 5 makes it challenging to comprehend. Please ensure that the references are arranged numerically and that the results are clearly divided and labeled according to the communication column or another category. Creating a taxonomy for the results would enhance clarity. I strongly recommend improving the information presented in the table and reviewing its layout to enhance the overall quality of the manuscript.

And finally, the font used in the reference section is not formatted properly.

Regards, 

Reviewer 2 Report

Comments and Suggestions for Authors

The paper is qualified for acceptance. 

Author Response

No comments

Round 3

Reviewer 1 Report

Comments and Suggestions for Authors

Dear Authors,

I want to take a moment to provide feedback on your recent manuscript submission titled "Wearable Solutions Using Physiological Signals for Stress Monitoring in Individuals with Autism Spectrum Disorder (ASD): A Systematic Literature Review." After a thorough review, I suggest a few improvements to enhance your manuscript. Since you are writing a systematic review, some adjustments must be made to achieve the expected standards.

  1. The keyword selection should be organized in a single table, with appropriate columns indicating the database source, the string used, and the results achieved.
  2. Figure 1 does not depict data extraction; rather, it illustrates the complete selection process by PRISMA guidelines. Data extraction refers to the details collected after reviewing the papers, including the publication year, location, research type, types of sensors used, and the hardware and software employed, among other factors. In qualitative research, data extraction involves gathering contributions from the papers to inform your analysis.
  3. The text in Figure 1 was highlighted in red by an automatic corrector. Please remove this highlighting.
  4. The outcomes of qualitative analysis are derived from the responses to the research questions. It is advisable to incorporate a comprehensive list of research questions aligned with the PICO framework within the methodology section. The methodology should adhere to the PICOS format, culminating in constructing the research string for database querying and articulating the research questions.
  5. In light of the outlined selection process, I strongly recommend establishing a dedicated section titled "Results." This section should comprise subsections that directly address each research question formulated in the methodology. This section is critical as it elucidates findings and highlights gaps identified throughout the study. Furthermore, it would be beneficial to incorporate graphical representations of the quantitative data derived from the 31 reviewed articles. The integration of visual tools will significantly enhance the readers’ comprehension of the findings presented in the journal.

Best regards,

Author Response

Comments 1: The keyword selection should be organized in a single table, with appropriate columns indicating the database source, the string used, and the results achieved.

Response 1: The keyword selection was changed in a sub-section titled “Search Strategy”

Comments 2: Figure 1 does not depict data extraction; rather, it illustrates the complete selection process by PRISMA guidelines. Data extraction refers to the details collected after reviewing the papers, including the publication year, location, research type, types of sensors used, and the hardware and software employed, among other factors. In qualitative research, data extraction involves gathering contributions from the papers to inform your analysis.

Response 2: It was modified (339-343 lines). Data extraction is related with Table 1.

Comments 3: The outcomes of qualitative analysis are derived from the responses to the research questions. It is advisable to incorporate a comprehensive list of research questions aligned with the PICO framework within the methodology section. The methodology should adhere to the PICOS format, culminating in constructing the research string for database querying and articulating the research questions.

Response 3: It was modified the order in the methodology and section news was created.

Comments 4: In light of the outlined selection process, I strongly recommend establishing a dedicated section titled "Results." This section should comprise subsections that directly address each research question formulated in the methodology. This section is critical as it elucidates findings and highlights gaps identified throughout the study. Furthermore, it would be beneficial to incorporate graphical representations of the quantitative data derived from the 31 reviewed articles. The integration of visual tools will significantly enhance the readers’ comprehension of the findings presented in the journal.

Response 4:  Section “Results” was changed sub-section “Extracted data” by Data Synthesis. Also, it was incorporated graphical representations of the quantitative data derived from the 32 reviewed articles.
